# Phosphoryl Trichloride–Method of Determination in Workplace Air

**DOI:** 10.3390/ijerph20136215

**Published:** 2023-06-24

**Authors:** Joanna Kowalska, Paweł Wasilewski

**Affiliations:** Central Institute for Labour Protection–National Research Institute, Czerniakowska 16, 00701 Warsaw, Poland; pawas@ciop.pl

**Keywords:** analytical chemistry method, workplace air, air quality, ion chromatography, occupational exposure

## Abstract

The aim of this study was to develop and validate a method for determining phosphoryl trichloride in workplace air. The method is based on passing the tested air through a sodium carbonate-impregnated quartz fiber sampling filter. The substance collected on the sampling filter is extracted with ultrapure water. Phosphoryl trichloride is determined as chloride ions (the product of the hydrolysis of phosphoryl trichloride) in the obtained aqueous solutions by ion chromatography with conductometric detection. The developed method enables the determination of phosphoryl trichloride in the air in the concentration range from 0.004 to 0.160 mg/m^3^. The method is not applicable in the presence of phosphorus trichloride, hydrochloric acid, and its salts in the air. Good validation results were obtained. All requirements of the norm PN-EN 482 were met while developing and validating the method. This method can be used to measure workplace air in order to assess workers’ occupational exposure.

## 1. Introduction

Phosphoryl trichloride (POCl_3_, CAS no. 10025-87-3) is a yellowish oily liquid that is volatile at room temperature. It hydrolyzes well in water with the formation of phosphoric and chloric acids; this reaction also takes place in humid air with the formation of chloric acid vapors; therefore, the vapors of the compound have corrosive properties and a pungent odor. In the European Union, phosphoryl trichloride is classified as a substance that is acutely toxic, harmful to organs, and causes skin irritation. The substance has been given the following hazard statements: H330: Fatal if inhaled, H372: Causes damage to organs, H302: Harmful if swallowed, and H314: Causes severe skin burns and eye damage [1]. Due to volatility, the main route of entry into the human body is through the respiratory tract. Symptoms of poisoning with phosphoryl trichloride are irritation and burns of the skin, eyes, and respiratory tract, shortness of breath, cough, pulmonary edema, headache, vomiting, and weakness [2].

Phosphoryl trichloride is widely used in industry, for example, in the synthesis of alkyl and aryl phosphates, carboxylic acid halogens, esters, also as an anhydrous solvent, as a solvent in cryoscopy; in the production of insecticides, plastics, elastomers, and pharmaceuticals [3,4,5,6]. It also serves as a precursor of war gases, and therefore its sale and export are strictly controlled [3].

In 2002, the estimated global production capacity for phosphoryl trichloride was 200,000 tonnes, with around 15 producers involved in its manufacturing. Among these, approximately 150,000 tonnes/year of the capacity was located in OECD countries, while the remaining 50,000 tonnes/year was distributed across non-member countries. In Western Europe, there were four producers of phosphoryl trichloride, three of which operated production plants in Germany [3].

In 2019 in Poland, at the meeting of the Expert Group on Chemical Factors of the Interdepartmental Commission on Maximum Permissible Concentrations and Intensity of Health Hazardous Factors in the Work Environment the value of 0.064 mg/m^3^ was proposed as the maximum allowable concentration (MAC) and 0.13 mg/m^3^ as the maximum allowable short-term concentration (MAC-STEL) for phosphoryl trichloride [7]. There is a need to develop a method for the determination of phosphoryl trichloride in the air in the range of 0.1 to 2 times its allowable concentration, which corresponds to the concentration from 0.0064 to 0.128 mg/m^3^.

The main objective of conducting an occupational risk assessment is to enhance working conditions and safeguard the well-being and health of workers. To effectively mitigate risks arising from physical, chemical, and biological hazards, it is crucial to possess reliable information regarding the extent of the hazard. The most accurate representation of a worker’s exposure to a harmful agent is derived from a numerical exposure index based on actual measurement results.

By comparing exposure rates with established maximum allowable concentrations and intensities of health-hazardous factors in the work environment, it becomes possible to evaluate the level of risk and make informed decisions regarding the implementation of corrective and/or preventive measures. This process ensures that appropriate actions are taken to manage and reduce risks, ultimately leading to improved occupational safety and the well-being of workers.

For phosphoryl trichloride, there are no standardized air monitoring methods available [4,8]. Difficulties in measuring the content of phosphoryl trichloride in the air result from its reactivity with atmospheric moisture.

Methods for the determination of similar chemical compounds, such as phosphorus trichloride, are available in the literature. NIOSH recommends a method involving spectrophotometric colorimetric analysis of the substance, obtained by oxidizing PCl_3_ and then carrying out the reaction using molybdenum salts. This method allows the determination of PCl_3_ at 1.2 mg/m^3^ for a 100 L air sample using a bubbler with 15 mL H_2_O [9].

POCl_3_ reacts during hydrolysis to hydrochloric acid, for which NIOSH recommends method 7907, which involves sampling HCl aerosol on a quartz filter impregnated with Na_2_CO_3_. This method allows the determination of HCl at 0.04 mg/m^3^ for a 600 L air sample. The analyte is then analyzed using ion chromatography [10].

The paper presents the methodology for the determination of phosphoryl trichloride in the air at the workplace, based on the conversion of POCl_3_ to hydrochloric acid and quantitative determination of chloride ions [11], which meets the requirements for the procedures for measuring chemical agents [12] and the legal requirements in Poland.

## 2. Materials and Methods

The following equipment was used in the studies: Dionex ICS-500 ion chromatograph (Dionex Corporation, Sunnyvale, CA, USA) equipped with Dionex AS-AP autosampler, ASRS 300 (4 mm) suppressor, and conductivity detector (Dionex Corporation, Sunnyvale, CA, USA). The tests were performed using a Dionex IonPac^®^AS22 anionic column (250 × 4 mm, 6 µm) with AG22 protective column (50 × 4 mm, 11 µm) (Dionex Corporation, Sunnyvale, CA, USA). The air sampler consisted of a three-piece cassette (SKC Inc., Eighty Four, PA, USA) with a sodium carbonate-impregnated quartz filter (Whatman, Maidstone, UK) of 37 mm diameter. A GilAir PLUS Personal Air Sampling Pump (Sensidyne, St. Petersburg, FL, USA) was used for air sampling. The SONIC-5 ultrasonic bath manufactured (Polsonic, Warsaw, Poland) was used to recover the analytes from the filters. In addition, the following were used: Teflon syringe filters (pore diameter 0.45 μm and diameter 25 mm) manufactured (Biosens, Warsaw, Poland). Additionally, the following reagents and materials were used: phosphoryl trichloride (Phosphorus(V) oxychloride) (Sigma-Aldrich, Darmstadt, Germany), inorganic anion reference solution Dionex™ Combined Seven Anion Standard II containing: F^−^ (20 mg/L), Cl^−^ (100 mg/L), NO_2_^−^ (100 mg/L), Br^−^ (100 mg/L), NO_3_^−^ (100 mg/L), PO_4_^3−^ (200 mg/L), SO_4_^2−^ (100 mg/L) (Thermo Fisher Scientific, Waltham, MA, USA), hexane (Sigma-Aldrich, Darmstadt, Germany), sodium carbonate, anhydrous, ≥99.5% (Sigma-Aldrich, Darmstadt, Germany), deionized (DI) water obtained from the Elix 3 system by Millipore (Merck Millipore, Burlington, VT, USA). Sodium bicarbonate and sodium carbonate mixture (NaHCO_3_/Na_2_CO_3_) prepared using a Dionex AS Eluent Concentrate NaHCO_3_/Na_2_CO_3_ (4.5 mM NaHCO_3_/1.4 mM Na_2_CO_3_) standard by Thermo Scientific (Thermo Fisher Scientific, Waltham, MA, USA) was the carrier phase for the IC analysis.

Air sampling was performed following the method described in norm ISO 21438-2 [13] and in Howe et al.’s article [14]. Single 37 mm diameter quartz filters were impregnated with 500 µL of 1 mol/l sodium carbonate solution and stored in a desiccator after drying. In order to collect POCl_3_ from the air, a system consisting of a battery-powered pump was used, which induces an airflow of 2 L per minute through an impregnated quartz fiber filter placed in the cassette.

The substances deposited on the sodium carbonate-impregned quartz filter were extracted with 10 mL of deionized water for 30 min in an ultrasonic bath. After standing (30 min), the solutions above the filters were passed through Teflon syringe filters.

Determination of chloride ions in aqueous solutions was performed by ion chromatography (IC) with conductometric detection [13]. The studies adopted the basic parameters described in previous studies [15]. The separation was performed on the Dionex™ IonPac™ AS22 column at a temperature of 30 °C. The carbonate/bicarbonate eluent was maintained at an isocratic flow rate of 1.2 mL/min. The standard calibration solutions have been covering the range of approximately 0.2 to 8.0 mg/L of chloride ions. Chlorides were separated from other co-occurring anions using these conditions.

## 3. Results and Discussion

### 3.1. Verification of the Correctness of the Adopted Method for the Determination of Phosphoryl Trichloride

The hydrolysis equation of phosphoryl trichloride: (1)$$POCl_{3}+H_{2}O\to H_{3}PO_{4}+3HCl$$

Due to the restrictions in the purchase of a pure phosphoryl trichloride standard, the risk posed by this substance to the health of laboratory workers and the ease of hydrolysis of this substance in humid air, it was assumed that POCl_3_ would be determined by the indirect method of determining chloride ions in aqueous solutions.

The samples prepared from POCl_3_ stock solutions in hexane, and DI water were evaluated for their degree of hydrolysis. For this purpose, POCl_3_ solutions with a concentration of 0.329 mg/mL were prepared by dissolving 5 µL of phosphoryl trichloride in (A) 25 mL of hexane and (B) in 25 mL of deionized water. Test solutions were prepared: A1 (20 µL of solution A in 5 mL of DI H_2_O), A2 (40 µL of solution A in 10 mL of DI H_2_O), B1 (20 µL of solution B in 5 mL of DI H_2_O) and B2 (40 µL of solution B in 10 mL of DI H_2_O). The concentration of POCl_3_ in the prepared solutions was 1.316 µg/mL, and according to the POCl_3_ hydrolysis reaction equation, the concentration of Cl^−^ in the tested solutions should be 0.913 µg/mL. The mean value of the concentration of the determined chloride ions (Table 1) was 0.949 µg/mL and differed from the value calculated from the hydrolysis reaction by 3.9% (this value is comparable with the value of the coefficients of variation obtained when determining the calibration curves and meets the presumed value requirements of less than 5%).

The obtained results indicate that the POCl_3_ hydrolysis reaction is 100% complete. Standard solutions of chloride ions can replace the use of phosphoryl trichloride in the determination method. The validation of the method was carried out with the use of standard solutions of chloride ions. It is crucial to bear in mind that any substances present in the tested air that have the potential to release chloride ions can lead to an overestimation of the determination results.

Due to the easy decomposition of POCl_3_ by reaction with water vapor in the contained air, there are no direct methods for determining this substance.

### 3.2. Recovery Efficiency Tests

The recovery from impregnated quartz filters for chloride ions at three concentration levels within the measuring range was determined. The filters (six samples for each concentration level) were loaded with 50, 300, and 400 µL of chloride ion solution (concentration 100 µg/mL). The following day, after the extraction with DI water, the solutions above the filters were subjected to chromatographic analysis.

For each level, also three reference solutions with appropriate concentrations were prepared: 0.5, 3.0, and 4.0 µg/mL. The obtained solutions were analyzed using a chromatograph. After reading the peak areas on the chromatograms of the tested solutions, the recovery was calculated (Table 2). The obtained mean recovery for chloride ions within the measuring range was 99.7%.

The recovery coefficient for phosphoryl trichloride from impregnated quartz filters was also determined in the same way. The following test was performed: 30 µL of a solution of phosphoryl trichloride in n-hexane (concentration 0.329 mg/mL) was applied to the prepared filters. The filters were then subjected to the same preparation process as in the recovery efficiency test.

Reference solutions were also prepared by adding 30 µL of a solution of phosphoryl trichloride in n-hexane to 3 flasks of 10 mL, which were filled with deionized water. After the chromatographic analysis of the tested solutions, the recovery was determined. The average recovery was 99.8%. This result is comparable to the chloride ion recovery from the filter. The obtained results validate the use of chloride ion standard solutions, eliminating the necessity for environmental laboratory personnel to acquire, store, and handle a hazardous chemical reagent like pure POCl_3_.

### 3.3. Verification of the Proposed Method of Air Sampling

The suitability of Na_2_CO_3_-impregnated quartz filters for air sampling was checked. For this purpose, two impregnated filters were placed in series and separated from each other in the 3-piece cassette. A total of 15 µL of a solution of phosphoryl trichloride in n-hexane (concentration 0.329 mg/mL) was applied to the first filter, and the air was passed through the cassette for 6 h (air flow rate was 2 L/min). After this time, the filters from the samplers were transferred separately to sealable polyethylene containers, each with 5 mL of deionized water, and subjected to ultrasonic waves for 30 min. After standing (30 min), the solutions above the filters were passed through syringe filters and then chromatographically analyzed.

The results of the chromatographic analysis showed that 98.6% of the determined chloride ions were in the solution from the first filter. After 720 L of air had passed through the cassette, only 1.4% of the substance was determined on the second filter. The test results confirm the validity of using impregnated quartz filters for air sampling.

### 3.4. Determining the Measuring Range and Calibration Tests

The standard calibration solutions ranged from 0.2 to 8 mg/L of chloride anions. Three series of calibration solutions were prepared. After chromatographic analysis of the calibration solutions, the dependence of Cl^−^ concentration (in µg/mL) on the peak area was plotted. The obtained model curves are described by the first-degree calibration function y = ax + b. The obtained value of the correlation coefficient was 0.9998. The analysis of the calibration curves was performed, and the Student’s *t*-test was used to check the significance of the intercept (for the number of degrees of freedom n-2 and the assumed probability level of 0.05). Intercepts for the determined calibration curves did not differ from the value of 0 in a statistically significant manner and can be omitted. Finally, the calibration curve took the form of y = 0.28x.

### 3.5. Validation Parameters of the Developed Method

All requirements of the PN-EN 482 norm were met while developing and validating the method [12]. The requirements for measuring methods for the determination of chemical agents in workplace air include precision, linearity, accuracy, quantification, sensitivity, and relative expanded uncertainty.

The calculated value of the expanded uncertainty (for the 95% confidence level) was 22% (long-term reference period) and 23% (short-term reference period). The limit of detection (LOD) was 15 ng Cl^−^ w 1 mL DI water, and the limit of quantification (LOQ value calculated as LOQ = 3·LOD) was 45 ng Cl^−^ w 1 mL.

### 3.6. Calculation of the Determination Results

The obtained mass concentration of chloride anion (µg/mL), having factored in the volume of deionized water used for recovery (mL), the recovery factor, the conversion factor (to convert concentration of Cl^−^ to POCl_3_) equal 1.4413, and the volume of the air sample (l), allows the calculation of phosphoryl trichloride concentrations in the analyzed air (mg/m^3^).

## 4. Conclusions

A method for determining phosphoryl trichloride (as chloride ions) using ion chromatography with conductivity detection in the workplace air for the assessment of occupational exposure was developed. The proposed chromatographic conditions allowed us to carry out the determination of chloride ions in the presence of other anions. However, the method is not applicable in the presence of phosphorus trichloride, hydrochloric acid, and its salts in the air. The developed method consists of the adsorption of phosphoryl trichloride on a quartz filter impregnated with sodium carbonate, extraction with deionized water, and analysis of the obtained solution. The range of the calibration curve corresponds to the POCl_3_ concentration range of 0.004 ÷ 0.160 mg/m^3^ (for the 720 L air sample and 10 mL of DI water for extraction).

Future research should focus on developing a method capable of determining POCl_3_ in the presence of hydrochloric acid, chlorides, or other substances that hydrolyze and generate chloride ions. One potential approach could involve derivatizing POCl_3_ into a stable derivative during the air sampling stage, thereby allowing for its subsequent quantification using a sensitive analytical technique. This direction of investigation holds the potential to overcome the challenges posed by the interfering substances and enhance the accuracy of POCl_3_ measurement in air samples.

## Figures and Tables

**Table 1 ijerph-20-06215-t001:** Determination of chloride ions in POCl_3_ solutions.

Solution	Peak Area of Cl^−^	Mean Peak Area of Cl^−^	Cl^−^ Concentration as Read from the Calibration Curve [µg/mL]	Coefficient of Variation for Concentration [%]
A1	0.2666	0.2666	0.952	0.5
	0.2665			
A2	0.2638	0.2642	0.944	
	0.2645			
B1	0.2680	0.2667	0.954	
	0.2654			
B2	0.2653	0.2648	0.946	
	0.2642			

**Table 2 ijerph-20-06215-t002:** Determination of Cl^−^ recovery from the sodium carbonate-impregnated quartz filter.

Cl^−^ Mass Applied onto the Filter [µg]	Area of Peaks in Recovery Solutions		Average Area of Peaks in Comparative Solutions	Recovery Factor		
	Average Value	Standard Deviation		Relative Standard Deviation [%]	Average Value	Relative Standard Deviation [%]
5.00	0.142	4.90	0.142	1.00	0.04	4.23
30	0.806	2.11	0.823	0.98	0.02	2.15
40	1.122	4.46	1.114	1.01	0.04	3.99

## Data Availability

Not applicable.

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
