# Peer review of "Phosphoryl Trichloride–Method of Determination in Workplace Air"

_ijerph, 2023, doi:10.3390/ijerph20136215_

Round 1
Reviewer 1 Report
Dear authors, thanks for valuable work.
In order to improve it, I suggest you next minor revisions:
1. Reprase sentences given in 56 - 60 line, as it seems quite "copied",
2. Improve the introduction part with additional reflection to occupational risk management in general, professional risk assessment, importance of physical, chemical and biological hazards monitoring,
3. Expand the reference section with the addition of several more references in the field of monitoring of similar chemical compounds, especially related to the definition of new measurement methods
Kind regards
Author Response
Thank you very much for your comments concerning the manuscript ijerph-2416048 entitled "Phosphoryl Trichloride – Method of determination in workplace air" submitted to International Journal of Environmental Research and Public Health.
Below you will find enclosed the response to the comments with the actions undertaken in order to modify the manuscript.
Reviewer: 1
Thank you very much for your insightful comments.
Ad. 1 „- Reprase sentences given in 56 - 60 line, as it seems quite "copied".”
- The sentence was modified.
Ad. 2 „- Improve the introduction part with additional reflection to occupational risk management in general, professional risk assessment, importance of physical, chemical and biological hazards monitoring.”
- Introduction part was modified according to comment
Ad. 3 „- Expand the reference section with the addition of several more references in the field of monitoring of similar chemical compounds, especially related to the definition of new measurement methods”
- References to methods for similar compouds were added.
Reviewer 2 Report
Reviewer’s comments:
1. The introduction section requires a more in-depth review of the literature about POCl3 and related chlorine compound detections.
2. There are no figures, neither in the preparation nor in the test procedures and results. They can better describe the proposal and method, so it is suggested to include figures.
3. In the results section, subsections 3.1 – 3.6 describe the procedure of the method. It seems as before the results, which are in section 3.7 only as two tables without any comment, analysis, discussion, or comparison with the cases of using other chlorine compounds that are explicitly indicated in the abstract (phosphorus trichloride, hydrochloric acid, and its salts in the air).
4. This is a short but incomplete manuscript about a proposed method for detecting POCl3. An appropriate discussion about Tables 1 and 2 and the results should be included. Otherwise, this is an incomplete technical report.

The text can be improved in its writing.
Author Response
Thank you very much for your comments concerning the manuscript ijerph-2416048 entitled "Phosphoryl Trichloride – Method of determination in workplace air" submitted to International Journal of Environmental Research and Public Health.
Below you will find enclosed the response to the comments with the actions undertaken in order to modify the manuscript.
Reviewer: 2
Thank you very much for your comments.
Ad. 1 „ The introduction section requires a more in-depth review of the literature about POCl3 and related chlorine compound detections.”
- References to methods for similar compouds were added,
- Introduction part was expanded with methods for similar compounds.
Ad. 2 „ There are no figures, neither in the preparation nor in the test procedures and results. They can better describe the proposal and method, so it is suggested to include figures.”
- In section 3.1 Fig. 1 was added containing hydrolysis reaction of POCl3,
- In section 2. more information about method conditions.
Ad. 3 & 4 „ In the results section, subsections 3.1 – 3.6 describe the procedure of the method. It seems as before the results, which are in section 3.7 only as two tables without any comment, analysis, discussion, or comparison with the cases of using other chlorine compounds that are explicitly indicated in the abstract (phosphorus trichloride, hydrochloric acid, and its salts in the air).”, “This is a short but incomplete manuscript about a proposed method for detecting POCl3. An appropriate discussion about Tables 1 and 2 and the results should be included. Otherwise, this is an incomplete technical report.”
- Tables 1 and 2 were moved and described in sections 3.1 and 3.2.
Reviewer 3 Report
In the introduction, it is important to justify the need to develop a method for phosphoryl trichloride determination, highlighting the health risks to workers and the importance of proper monitoring in the work environment. In addition, it is essential to discuss the wide use of this compound in industry, mentioning specific examples of application and providing scientific references to support the information presented.
In materials and methods, it is essential to provide specific details about the elution conditions, solvent gradient, and other parameters relevant to the separation and detection of phosphorus trichloride. In addition, it is necessary to clearly explain the procedure for impregnating the quartz filter with sodium carbonate, the process for collecting the phosphorus trichloride, and the conditions for extracting the analytes from the quartz filter. It is also important to include information about the ion chromatography system, such as column type, electrolyte concentration, eluent flow rate, and temperature.
In the results section, I suggest you justify the choice of the indirect method and perform a comparison with the direct methods available in the literature to support this choice. In addition, it is relevant to establish acceptance criteria for the accuracy of the method and evaluate whether the difference of 3.9% is within acceptable limits. A more detailed analysis of the variation between samples in the recovery efficiency tests, with the calculation of mean values and variability of recoveries at each concentration level, is also suggested.
In the discussion, it is essential to provide further analysis of the results, relating them to existing literature and discussing their implications. In addition, it is important to highlight the limitations of the study and suggest possible future improvements to the methodology.
In conclusion, it is necessary to reassess the synthesis of the main findings and highlight the specific contribution of the study to scientific knowledge in the field. It is also relevant to mention the limitations of the study and suggest directions for future research.
I thank you for your attention to these suggestions and remain at your disposal for further clarification if needed.
There is a need for English revision due to possible misinterpretations.
Author Response
Thank you very much for your comments concerning the manuscript ijerph-2416048 entitled "Phosphoryl Trichloride – Method of determination in workplace air" submitted to International Journal of Environmental Research and Public Health.
Below you will find enclosed the response to the comments with the actions undertaken in order to modify the manuscript.
Reviewer: 3
Ad. „ In the introduction, it is important to justify the need to develop a method for phosphoryl trichloride determination, highlighting the health risks to workers and the importance of proper monitoring in the work environment. In addition, it is essential to discuss the wide use of this compound in industry, mentioning specific examples of application and providing scientific references to support the information presented.”
- Introduction part was modified according to comment.
Ad. „ In materials and methods, it is essential to provide specific details about the elution conditions, solvent gradient, and other parameters relevant to the separation and detection of phosphorus trichloride. In addition, it is necessary to clearly explain the procedure for impregnating the quartz filter with sodium carbonate, the process for collecting the phosphorus trichloride, and the conditions for extracting the analytes from the quartz filter. It is also important to include information about the ion chromatography system, such as column type, electrolyte concentration, eluent flow rate, and temperature.”
- Materials and method parts were changed according to comment.
Ad. „ In the results section, I suggest you justify the choice of the indirect method and perform a comparison with the direct methods available in the literature to support this choice. In addition, it is relevant to establish acceptance criteria for the accuracy of the method and evaluate whether the difference of 3.9% is within acceptable limits. A more detailed analysis of the variation between samples in the recovery efficiency tests, with the calculation of mean values and variability of recoveries at each concentration level, is also suggested.”
- Result section was modified according to comment,
- Accuracy acceptance criteria was added.
Ad. „ In the results section, I suggest you justify the choice of the indirect method and perform a comparison with the direct methods available in the literature to support this choice. In addition, it is relevant to establish acceptance criteria for the accuracy of the method and evaluate whether the difference of 3.9% is within acceptable limits. A more detailed analysis of the variation between samples in the recovery efficiency tests, with the calculation of mean values and variability of recoveries at each concentration level, is also suggested.”
- Discussion was provided.
Ad. „ In conclusion, it is necessary to reassess the synthesis of the main findings and highlight the specific contribution of the study to scientific knowledge in the field. It is also relevant to mention the limitations of the study and suggest directions for future research.”
- Limitations of the study and directions for future research were added.
Round 2
Reviewer 2 Report
The changes in the manuscript are adequate and improve the quality of the work.
The Quality of English is adequate.
Reviewer 3 Report
Dear editor,
The authors made the suggested corrections and formally accepted the study for publication.
Thanks for the opportunity.
My greetings.
Dear editor,
The authors made the suggested corrections and formally accepted the study for publication.
Thanks for the opportunity.
My greetings.